# Delaying an Electromagnetic Pulse with a Reflective High-Integration Meta-Platform

**DOI:** 10.3390/nano14171438

**Published:** 2024-09-03

**Authors:** Liangwei Li, Weikang Pan, Yingying Wang, Xiangyu Jin, Yizhen Chen, Zhiyan Zhu, Muhan Liu, Jianru Li, Yang Shi, Haodong Li, Shaojie Ma, Qiong He, Lei Zhou, Shulin Sun

**Affiliations:** 1Shanghai Engineering Research Centre of Ultra Precision Optical Manufacturing, Department of Optical Science and Engineering, School of Information Science and Technology, Fudan University, Shanghai 200433, China; 21210720008@m.fudan.edu.cn (L.L.); 19110720004@fudan.edu.cn (W.P.); 21210720013@m.fudan.edu.cn (Y.W.); 22210720009@m.fudan.edu.cn (X.J.); chenyz@lut.edu.cn (Y.C.); 21110720021@m.fudan.edu.cn (Z.Z.); 21110720013@m.fudan.edu.cn (M.L.); 23110720008@m.fudan.edu.cn (J.L.); 22110720011@m.fudan.edu.cn (Y.S.); shaojiema@fudan.edu.cn (S.M.); 2Yiwu Research Institute, Fudan University, Chengbei Road, Yiwu 322000, China; 3School of Science, Lanzhou University of Technology, Lanzhou 730050, China; 4State Key Laboratory of Surface Physics (Ministry of Education), Fudan University, Shanghai 200433, China; 19110190007@fudan.edu.cn (H.L.); qionghe@fudan.edu.cn (Q.H.); 5Key Laboratory of Micro and Nano Photonic Structures (Ministry of Education), Fudan University, Shanghai 200433, China; 6Collaborative Innovation Center of Advanced Microstructures, Nanjing 210093, China

**Keywords:** meta-surface, time delay, Pancharatnam–Berry phase, dispersion engineering

## Abstract

Delaying an electromagnetic (EM) wave pulse on a thin screen for a significant time before releasing it is highly desired in many applications, such as optical camouflage, information storage, and wave–matter interaction boosting. However, available approaches to achieve this goal either require thick and complex systems or suffer from low efficiencies and a short delay time. This paper proposes an ultra-thin meta-platform that can significantly delay an EM-wave pulse after reflection. Specifically, our meta-platform consists of three meta-surfaces integrated together, of which two are responsible for efficiently coupling incident EM-wave pulse into surface waves (SWs) and vice versa, and the third one supports SWs exhibiting significantly reduced group velocity. We employ theoretical model analyses, full-wave simulations, and microwave experiments to validate the proposed concept. Our experiments demonstrate a 13 ns delay of an EM pulse centered at 12.975 GHz, enabled by a λ/8-thick and 38-λ-long meta-device with an efficiency of 32% (or 70%) with (or without) material loss taken into account. A larger delay time can be enabled by devices with larger sizes considering that the SWs group velocity of our device can be further reduced via dispersion engineering. These findings establish a new road for delaying an EM-wave pulse with ultra-thin screens, which may lead to many promising applications in integration optics.

## 1. Introduction

Electromagnetic (EM) waves are widely used to probe the properties of matter and convey information; however, the interaction time between EM waves and matter is usually too short due to the high speed of EM waves, which is unfavorable for these applications. Therefore, a thin screen capable of trapping an EM pulse for a significantly long time is highly desired in diverse fields, such as optical storage and processing [1,2], enhanced wave–matter interactions [3,4,5,6], and optical camouflage [7].

Conventional slow-wave devices are usually complex structures of wavelength-scale thicknesses. Intuitively, the delay time of an EM-wave pulse enabled by a device can be roughly described by $\tau_{g}=L/v_{g}$, where L denotes the effective distance traveled by the pulse and $v_{g}$ represents the group velocity of light at the center frequency of the pulse. Therefore, $\tau_{g}$ can be enlarged by increasing L through utilizing spiral waveguides [8,9] and optical fibers with long lengths [10,11]. While such methods can delay the light at the picosecond (ps) to nanosecond (ns) level, they still encounter inherent problems including a large spatial footprint and high propagation loss. Alternatively, one can design thin devices supporting significantly reduced $v_{g}$, based on dispersion engineering of the refractive index and band structure, such as coherent population oscillations [12], electromagnetically induced-transparency (EIT) resonance [13,14] and its classical counterpart [15,16,17,18,19,20], and photonic crystal and grating waveguides [21,22,23,24,25,26]. However, these schemes usually exhibit strong frequency dispersions, leading to a narrow working bandwidth and time-domain distortions on the incident pulse.

Meta-surfaces, two-dimensional meta-materials composed of subwavelength microstructures (e.g., meta-atoms) exhibiting spatially tailored scattering properties, have recently exhibited extraordinary capabilities to control EM waves [27]. Designing appropriate meta-atoms or choosing appropriate global sequences, people have developed different meta-surfaces capable of realizing many fascinating wave-manipulation effects such as anomalous light bending [28,29], propagating waves (PWs) to surface waves (SWs) conversion [30,31,32,33], meta-lensing [34,35,36], EM cloaking [37,38], and holograms [39,40,41], etc. Moreover, meta-surfaces have also been designed to enable slow-wave transportation of SWs or even trapping of SWs [42,43,44]. These devices are usually of deep-subwavelength thicknesses, which is highly compatible with on-chip applications. Nevertheless, to the best of our knowledge, meta-surfaces capable of delaying an incident PW pulse (rather than a SW pulse) have not been realized.

In this article, we propose an ultra-thin and high-efficiency on-chip meta-device, consisting of three different meta-surfaces integrated together, to trap an incident PW pulse for a significant time before releasing it back to free space. Specifically, two meta-surfaces are the meta-coupler and meta-decoupler, responsible for efficiently converting incident free-space PWs into on-chip SWs and eventually decoupling SWs back to PWs, respectively. In particular, a slow-wave meta-surface, located between these two devices, is utilized to gradually slow down and then speed up the SWs traveling along the on-chip space through dispersion engineering (see Figure 1). We first employ effective-medium model analyses to validate the proposed concept, and then fabricate a microwave sample with a λ/8 thickness and experimentally demonstrate that it can delay a PWs pulse with a center frequency of 12.975 GHz for 13 ns before reflecting it back to free space. Here, the group velocity of SWs is reduced to 0.08 c and the efficiency of our device is 32% (or 70%) with (or without) material loss taken into account. Considering that the group velocity of SW can be further reduced, the delay time will be further increased by enlarging the lateral dimension of the device.

## 2. New Concept for Delaying an EM Pulse by Integrated Meta-Device

We begin by introducing our approach for delaying EM waves using an integrated meta-surface, as illustrated in Figure 1a. It is noted that, when EM waves are impinged on a conventional interface such as a flat metallic film, they are directly reflected with almost no delay time. Therefore, it offers negligible possibility to temporally trap the light for a while and enhance its interaction with the material. The proposed integrated meta-device, comprising a meta-coupler, a slow-wave meta-surface, and a meta-decoupler, achieves the delay of light based on two primary mechanisms: mode conversion and velocity manipulation of light. As depicted in Figure 1a, the first meta-coupler efficiently couples the incident PWs to SWs traveling along the device by utilizing the encoded phase gradient for compensating the wave-vector gap between these two modes. Following that, another slow-wave meta-surface can slowly reduce and then release the velocity of SWs traveling on them through inhomogeneous dispersion engineering. Finally, a meta-decoupler releases the on-chip SWs back to free-space PWs, thereby achieving the time delay effect for input waves. Distinct from conventional systems, such an integrated meta-device is capable of achieving a significant delay time for free space light by fully utilizing both the enlarged space and the decreased velocity of SWs on the chip.

To demonstrate the feasibility of our scheme, we design a meta-device based on an effective media model and perform numerical simulations to observe the desired time delay effect of EM waves (Simulation settings can be found in Appendix B). The proposed device is constructed by covering a flat metallic film with an ultrathin dielectric layer, which has a spatially varying refractive index. For example, we design the meta-coupler with the dielectric parameters described as follows [30] (see Figure 1b):(1)$$\varepsilon_{m1}\left(x\right)=\mu_{m1}\left(x\right)=1+\frac{\xi x}{2d_{m}k_{0}}$$
where ξ is the phase gradient of the meta-coupler; $d_{m}$ is the dielectric thickness; $k_{0}$ is the wave-vector in vacuum. Such a meta-coupler, possessing a reflection phase slope exceeding the total wave-vector of light, can convert PWs to SWs with nearly perfect efficiency [32]. Without loss of generality, we choose the following set of parameters: $\xi =1.23k_{0}$, $d_{m}=1\;mm$, and $f_{0}=12.975\;GHz$. To prevent the dielectric parameters from becoming unreasonably large, we introduce a periodic condition to truncate their values in the theoretical model. After guiding the input PWs into the on-chip space, we further adopt a slow-wave meta-surface to control the velocity of the excited SWs. Designed in the same configuration, such a meta-surface allows guided SWs to gradually slow down by introducing a gradient refraction index distribution for the top dielectric layer as shown in Figure 1c. Here, the dielectric parameters of the slow-wave meta-surface are set to be $\varepsilon_{m2}\left(x\right)=\mu_{m2}\left(x\right)=25$ at the center and $\varepsilon_{m2}\left(x\right)=\mu_{m2}\left(x\right)=2.8$ at the two edges, creating a trapezoid-like distribution. It should be noted that the high refractive index will bend down the dispersion relation of SWs, giving rise to an extremely slow group velocity at the central area of the meta-surface. Furthermore, the gradient parameter distribution of the slow-wave meta-surface helps to minimize both reflection and scattering losses through gradual impedance matching. Finite-difference time-domain (FDTD) calculations, as shown in Figure 1c, reveal that the group velocity of the SW is 0.02 c and 0.6 c in the center and edge areas of the slow-wave meta-surface, respectively.

The final step in the scheme involves decoupling the SWs back to PWs, eventually achieving a large delay time for the impinging waves. Inspired by the recent advances in SWs meta-surfaces [45], we design a meta-decoupler to scatter the near-field SWs to free space PWs through an inverse wave-vector compensation. In this process, the radiation direction $\theta_{r}$ of the outgoing wave can be obtained from the following equation, i.e., $k_{0}sin\theta_{r}=k_{sw}+\xi \prime$, according to the wave-vector conservation of EM waves along the in-plane direction. The dielectric parameters of the meta-decoupler can be described in the following form (see Figure 1d):(2)$$\varepsilon_{m3}\left(x\right)=\mu_{m3}\left(x\right)=1+\frac{\pi }{{d_{m}k}_{0}}-\frac{\xi x}{2d_{m}k_{0}}$$

Next, we conduct a full-wave simulation to demonstrate the EM wave delay phenomenon of the effective-medium-based meta-device. As depicted in Figure 1e, a normally incident continuous Gaussian beam (waist width $w_{0}=50\;mm$; frequency $f_{0}=12.975\;GHz$) is converted to SWs by the left-side meta-coupler. Next, the SWs propagate on the slow-wave meta-surface with its group velocity gradually reduced down to 0.02 c and then released back to the initial value of 0.6 c. Ultimately, the SWs are reconverted into PWs along $\theta_{r}=0^{{}^{\circ}}$ by the right-side meta-decoupler (considering that ξ′ = $-\xi =-k_{sw}$). After completing the whole process, the outgoing EMs obtain a significant delay time of 36 ns, which can be further increased by enlarging the lateral length or increasing the refractive index of the slow-wave meta-surface. For comparison, at the interface between air and conventional devices, e.g., metallic film, the delay time obtained by the reflection beam nearly equals to zero. In addition, the working efficiency, defined as the energy ratio between the outgoing PWs and impinging PWs, reaches a value of 73% according to the numerical calculations. If continuously suppressing the reflection and scattering losses through the optimization of structural configurations, the performance of such an integrated meta-device can be further improved. To illustrate the whole process of the slow wave more clearly, we also adopt a transient simulation to observe the evolution of the EM pulse slowed down on our meta-device at different times. (See Appendix A).

## 3. Designs of the Integrated Meta-Device for Delaying an EM Pulse

After demonstrating our concept using an effective medium model, we proceed to design a realistic meta-device for slowing EM waves. We propose to adopt the Pancharatnam–Berry (PB) meta-coupler for coupling and decoupling SWs, which possesses the advantages of high efficiency, easy fabrication and integration, and so on [46,47]. For a generic reflection-typed PB meta-atom satisfying mirror symmetry, its reflection properties are represented by the Jones matrix $R=\left(\begin{array}{cc} r_{uu} & 0\\ 0 & r_{vv} \end{array}\right)$, in which $r_{uu}$ and $r_{vv}$ denote the co-polarization reflection coefficients for linear polarization (LP) along two principal axes u and v. Based on our previous works [48,49], such meta-atoms with an orientation angle θ illuminated by circular polarization (CP) light will generate two components, including a spin-conserved normal mode ${\tilde{r}}_{n}=\frac{1}{2}\left(r_{uu}+r_{vv}\right)$ and a spin-flipped abnormal mode ${\tilde{r}}_{a}=\frac{1}{2}e^{i\sigma \cdot 2\theta }\left(r_{uu}-r_{vv}\right)$, where σ=+1 or −1 indicates left-hand circular polarization (LCP) or right-hand circular polarization (RCP), respectively. It should be noted that only the abnormal mode acquires a spin-dependent geometric phase of ±2θ, which can be exploited to precisely tailor the wavefront of the light using the gradient PB meta-device. In contrast, the first term, which is entirely independent of the meta-atoms’ orientation, generates only the normal specular reflection mode, thus degrading the performance of the PB meta-device.

To achieve 100% efficiency, the desired PB meta-atoms need to completely eliminate the useless normal mode, giving rise to the criterion of $r_{uu}+r_{vv}=0$. Considering that the reflective meta-atoms will block the transmission channel entirely in a lossless device (i.e., ${|r}_{uu}|=\left|r_{vv}\right|=1$), we can re-describe the criterion as $\Phi_{uu}-\Phi_{vv}=\pi$, implying that the PB meta-atoms should function as an effective half-wave plate. We employ the polarization conversion ratio (PCR), defined as the intensity of anomalous mode ${\left|(r_{uu}-r_{vv})/2\right|}^{2},$ to evaluate the efficiency of the PB meta-atoms. As depicted in Figure 2a, the realistic PB meta-atoms are designed in a metal–insulator–metal (MIM) configuration, consisting of an H-shaped copper microstructure and a flat copper film separated by a dielectric spacer ($\varepsilon_{r}=3)$. Some detailed structural parameters of the PB meta-atoms are listed as follows: $w=0.5\;mm,\;t_{1}=4\;mm,\;t_{2}=2\;mm,\;h=3\;mm,\;p=6\;mm$. We fabricate a sample constituted by a periodic array of PB meta-atoms using the printed circuit board (PCB) technique, as illustrated in Figure 2a. In the experiments, we detect the co-polarization reflection phases $\phi_{uu}$ and $\phi_{vv}$ of the sample using two LP antennas as the source and detector, with their polarizations both aligned along the u and v axes, respectively. Figure 2b shows that the phase difference between $\phi_{uu}$ and $\phi_{vv}$ approximates 180° within the frequency range of 11–14 GHz. Since the reflection-typed meta-atoms satisfy ${|r}_{uu}|=\left|r_{vv}\right|=1$, the PCR of the sample remains at about 100%, demonstrating the equivalent behavior of the half-wave plate. FDTD simulation results (line) are consistent with the experimental data (dots), justifying that the created building blocks agree well with the theoretical criterion.

Now, we introduce the design of the building blocks of the proposed slow-wave meta-surface that can support the eigen SWs with well-controlled group velocity via structural tuning. It should be noted that, in microwave domains where natural SPPs do not exist, spoof SPPs based on structured metals [30,32,50,51,52] are utilized in the present study. The unit cell of the slow-wave meta-surface, shown in the inset of Figure 2c, is composed of a copper patch, a dielectric spacer, and a copper film arranged from top to bottom. Using full-wave simulations, we obtain the dispersion relation of the SWs supported by these periodic meta-atoms, as shown in Figure 2d. At the frequency of 12.975 GHz, the meta-atoms with patch width l = 0 mm support the eigen SWs with $k_{sw}=335\;m^{-1}$. As the width of the top copper patch increases from 0 mm to 3.5 mm, the slope of dispersion relation for the eigen SWs gradually decreases, implying that the group velocity of near-field light is slowed down. In particular, due to periodic scattering, the group velocity is close to zero around the Brillouin zone boundary. For example, as l = 3.5 mm, the group velocity of SWs is reduced to approximately 0.08 c at 12.975 GHz. Due to the limited space of the main text, the working mechanism and design details of the slow-wave meta-surface are summarized in Appendix A.

We next construct the meta-coupler that can efficiently convert input PWs to SWs. As shown in Figure 3a, such a device is composed of a series of identical PB meta-atoms with their orientations rotated clockwise at a constant step of $∆\varphi ={-58}^{{}^{\circ}}$. According to the mechanism of the geometric phase, the meta-coupler is able to provide a dispersionless phase gradient $\xi =335\;m^{-1}$ for impinging the RCP wave. We connect the meta-coupler to the slow-wave meta-surface with the initial value of l set as 0 mm, which can support the SWs with the eigen wave-vector of $k_{sw}=335\;m^{-1}$ at 12.975 GHz. FDTD simulations depicted in Figure 3c demonstrate that the normally incident RCP PWs can be efficiently coupled to SWs at the desired frequency, with their wave-vector gap compensated by the phase gradient of the meta-coupler. The PW-SW conversion efficiency, defined as the ratio of the power of outgoing SWs to incident PWs, reaches the high value of 80% at 12.975 GHz according to our calculations. The high performance can be attributed to the high PCR of PB meta-atoms and the wave-vector matching of our device.

Based on the mechanism of the inverse wave-vector compensation, we have successfully designed a meta-decoupler that can scatter the near-field SWs to the far-field PWs with high efficiency. According to our previous research [45], we construct a PB meta-decoupler composed of the identical PB meta-atoms (as depicted in Figure 2a) with their orientations rotated counter-clockwise with a constant step of $∆\varphi =58^{{}^{\circ}}$, as depicted in Figure 3b. As the impinging SWs interact with the PB meta-decoupler, they will acquire an opposite wave-vector compensation along the −x direction and, therefore, radiate to free space as PWs. In this process, the parallel wave-vector of the EM waves is conserved, i.e., $k_{0}sin\theta_{r}=k_{sw}+\xi^{\prime }$ [45]. It is clear that as $\xi^{\prime }$=$-335\;m^{-1}$ equals to $-k_{sw}$, the outgoing waves will be the PWs propagating along $\theta_{r}=0^{{}^{\circ}}$ (i.e., along the z axis). FDTD simulations demonstrate that the efficiency of the meta-decoupler, defined as the ratio of the power carried by the radiative PWs to the incident SWs, reaches an impressive value of 95% at 12.975 GHz.

To achieve a substantial delay time, reducing the velocity of the SWs based on our slow-wave meta-surface is crucial. Utilizing the unit cell depicted in Figure 2c, we have created a slow-wave meta-surface by integrating unit structures with two distinct patch widths (3.5 mm at the central area and 0 mm at the edge area), as illustrated in Appendix A. This configuration slows down the SWs significantly over a long on-chip distance, enabling a large delay time. However, there are strong reflection and scattering losses owing to a notable impedance mismatch at the boundary of the two different areas, leading to degraded performance of the slow-wave meta-surface [53] (see Appendix A). To address these issues, we have introduced a gradient design into the slow-wave meta-surface by progressively altering the patch width along the lateral space. This allows a smooth change, rather than an abrupt change, in the wave-vector (or impedance) of SWs traveling across the device [54,55]. Based on this strategy, the performance of the slow-wave meta-surface can be significantly improved through deeply suppressing both scattering and reflection effects. (See Appendix A).

## 4. Characterization of EM Pulse Delay Based on the Integrated Meta-Device

We fabricate the integrated meta-device through combining three elements presented previously, i.e., the meta-coupler, the slow-wave meta-surface, and the meta-decoupler. In our experiments, we adopt a near-field scanning measurement to characterize the delay effect of the EM pulse, with the experimental setup depicted in Figure 4a. In our measurements, a horn antenna, placed about 10 cm above the meta-coupler, illuminates the RCP EM pulse directly onto the device, while another monopole antenna, controlled by a motorized translation stage, is utilized to map the electric field distribution of the EM waves. Pyramid-shaped absorbing foams are placed at the boundary of the fabricated sample in order to absorb the outgoing EM waves or the surrounding EM noise. The results depicted in Figure 4b reveal that a normally incident Gaussian beam at the frequency of 12.975 GHz is converted to the SWs with the group velocity gradually reduced, ultimately being radiated back to free space based on the proposed meta-device. Moreover, FDTD simulations are further performed to evaluate the performance of the sample, demonstrating that the conversion efficiency is about 32% in the lossy case with $\varepsilon_{r}=3+0.002i$ (or 70% in the lossless case with $\varepsilon_{r}=3$). In this process, the reflection and scattering losses of generated SWs on the slow-wave meta-surface are extremely small, attributing to the mechanism of gradual impedance matching. The consistency between the near-field measurement and the FDTD simulations undoubtedly demonstrates the high performance of our meta-device.

To further validate our concept, we conduct the transient simulations based on the FDTD method to observe the evolution of the EM pulse inside our device. Figure 5 presents the E-field (E_x_) distribution of the EM waves traveling on our meta-device at different times. As shown in Figure 5a, a Gaussian beam pulse with a central frequency of 12.975 GHz is excited at the upper left-side boundary with the electric field described as follows:(3)$$\vec{E}=E_{0}exp\left[i\left(-k_{0}z-\omega t\right)\right]exp[-{\left(\frac{x-x_{0}}{w_{0}}\right)}^{2}]exp[-{\left(\frac{t-t_{0}}{T}\right)}^{2}]\hat{x}$$
where ω is the angular frequency; and $w_{0}$ is the beam waist width of the EM pulse. In the simulation, we assume $\omega =2\pi f_{0}$, $f_{0}=12.975\;GHz$, $w_{0}=50\;mm$, $x_{0}=75\;mm,$ $t_{0}=18\;ns$, T=10 ns. Considering the velocity of SWs on the slow-meta-surface can be extremely slow only at the band edge frequency, we should excite an EM pulse for quite a long length of time, ensuring that the beam carries a quasi-single frequency of 12.975 GHz. It is noted that the input beam is efficiently converted to the SWs by the meta-coupler with an initial group velocity of 0.61 c according to the dispersion relation of the SWs shown in Figure 2d. After encountering the slow-wave meta-surface, the SWs are gradually slowed down to 0.08 c and then released back to the initial value of 0.61 c through the gradient design strategy of the SW dispersion relations at different spatial positions. While an extremely large time delay is accumulated, the scattering and reflection effects of SWs are significantly suppressed during this process (see Figure 5b–d). Thereafter, at 34 ns, the SWs are successfully decoupled backed to free space propagating along the z axis by the meta-decoupler. Overall, the input EM pulse experiences a time delay of approximately 13 ns after reflection (see the discussion in the following paragraph), demonstrating the feasibility of the proposed meta-device-based slow-wave technology. For comparison, at the interface between air and conventional devices, e.g., metallic film, the delay time obtained by reflection beam nearly equals to zero. In addition, for an integrated meta-device without the slow-wave meta-surface, the accumulated delay time cannot be so large (see more discussion in Appendix A).

Additionally, we conduct far-field measurements to quantitatively evaluate the performance of our meta-device using a far-field measurement system depicted in Figure 6a. While the source antenna illuminates the quasi-single frequency EM pulse on the left-side meta-coupler, the receiver antenna collects the outgoing EM pulse emitted by the right-side meta-decoupler. Both antennas, placed at 10 cm above the sample, are connected to the vector network analyzer. Relying on such a setup, we can detect the time-domain signal of the EM beam traveling inside the device. For comparison, the input signal is obtained by placing the receiver antenna directly in front of the source antenna with a separation distance of 20 cm. Via measuring the interval between input and output EM pulses, we can determine the delay time of the light beam achieved by the integrated meta-device. We have conducted several experiments by launching EM pulses with different central frequencies. For example, with a central frequency of 12.5 GHz, the input beam undergoes the whole conversion process of PW-SW-PW by our meta-device, resulting in a total delay time of 5.6 ns, as shown in Figure 6b (the difference between measured and simulated spectra may be attributed to imperfections in our measurement setup and instability of our instruments). In this case, since the velocity of the SWs can decrease to only 0.4 c on the central region of the slow-wave meta-surface, the delay time is thus relatively small. For comparison, as the central frequency of the EM pulse is changed to 12.975 GHz, 12.98 GHz, and 12.985 GHz, the measured delaying time is 13 ns, 16 ns, and 20 ns, respectively. Since the frequency gradually approaches the band edge, the accumulated delay time thus continuously increases. For instance, at 12.975 GHz and 12.985 GHz, the group velocity of SWs at the center of the slow-wave meta-surface reduces to 0.08 c and 0.06 c. The integrated meta-device exhibits a large delay effect (at the tens of nanoseconds level) for impinging EM pulses at various operating frequencies, thanks to the two controllable degrees of freedom, i.e., both the propagation length and group velocity on the chip.

It should be noted that, as the frequency increases from 12.5 GHz to 12.985 GHz, the delay time improves from 5 ns to 20 ns, exhibiting a clear frequency-dependent behavior present in all strong-dispersion systems. Meanwhile, the efficiency of such a slow-wave device is sensitive to the working frequency (decreasing from 71% at 12.5 GHz to 9% at 12.985 GHz, see more details in Appendix A), because the light–matter interaction (i.e., the energy dissipation) is significantly enhanced around the band edge frequency. Figure 6c–e demonstrate that the output pulses exhibit a longer tail, attributed to the strong dispersion effect of SW around the band edge frequency. Although such strong frequency dependence is undesired for many applications, it may become beneficial in some scenarios, e.g., optical sensing and camouflage.

## 5. Conclusions

In summary, we propose a new strategy to achieve a large delay time of light based on a high-integration meta-platform with subwavelength thickness. Our device is composed of three components, including a meta-coupler for converting input PWs to SWs, a slow-wave meta-surface for accumulating a substantial delay time by extending the propagation length of SWs with slowed-down group velocity on the chip, and a meta-decoupler for radiating SWs back to free space PWs. A meta-device based on an ideal effective medium model is first investigated, exhibiting a large delay time of 36 ns and a high conversion efficiency of 73%. As a practical realization of our concept, we design and fabricate a realistic meta-device with only λ/8 thickness working at the microwave regime and experimentally demonstrated that a CP EM pulse with the central frequency of 12.975 GHz obtains a delay time of 13 ns with the group velocity of SWs slowed down to 0.08 c. We should emphasize that the delay time can be improved (e.g., 36 ns realized in the ideal effective medium model or beyond) by further increasing the lateral space and decreasing the SW velocity on the meta-device. In addition, this idea can also work for the LP light based on the resonant-phase meta-device. Compared to previous proposals, our scheme exhibits advantages such as a large delay time, ease of fabrication/integration, and subwavelength scale. We need to mention that the concept is quite general and this work provides an ideal platform for enhancing wave–matter interactions in microwave and high-frequency regions (see numerical demonstrations in Appendix A), which can be applied in diverse fields such as nonlinear enhancement, biosensing, and photonic storage.

## Figures and Tables

**Figure 1 nanomaterials-14-01438-f001:**
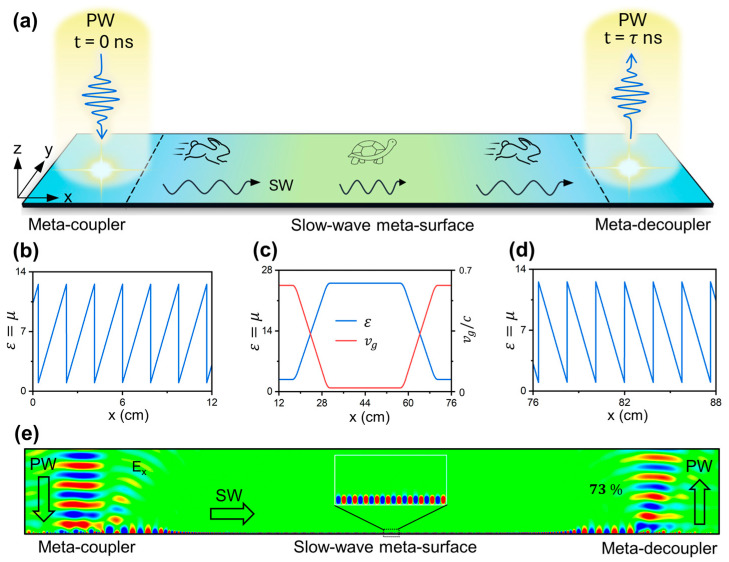
Schematic illustration and numerical demonstration of the effective-medium-based slow-wave meta-device. (**a**) Schematic of the slow-wave device consisting of a meta-coupler for PW-SW conversion, a slow-wave meta-surface for slowing down the speed of SW, and a meta-decoupler for SW-PW decoupling. (**b**–**d**) Dielectric parameters of the meta-coupler, slow-wave meta-surface, and meta-decoupler. Here, red line in (**c**) denotes the spatial distribution of group velocity of SW propagating along the slow-wave meta-surface. (**e**) Simulated E-field (Re(Ex)) distribution inside the integrated meta-device while a Gaussian beam is normally illuminated on the left-side meta-coupler at 12.975 GHz. According to the calculation, the effective-medium-based meta-device successfully achieves a large delay time of 36 ns for impinging light with a high efficiency of 73%. Here, the material loss is not included to make the device simple and the underlying physics clear.

**Figure 2 nanomaterials-14-01438-f002:**
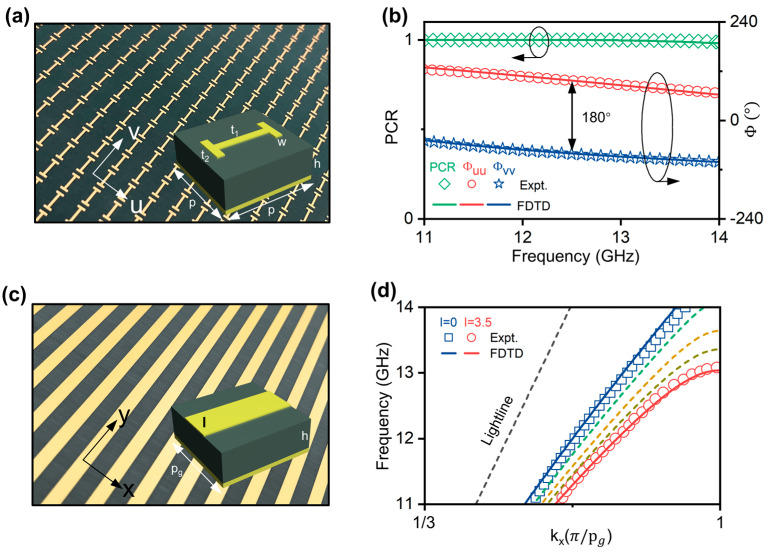
Characterization of the proposed building blocks for designing the integrated meta-device. (**a**) Sample picture of designed PB meta-atoms for building the meta-coupler and meta-decoupler. Some key geometric parameters are listed as follows: $w=0.5\;mm,\;t_{1}=4\;mm,\;t_{2}=2\;mm,\;h=3\;mm,\;p=6\;mm$. (**b**) Spectra of the co-polarization reflection phases ($\Phi_{uu}$ and $\Phi_{vv}$) and the polarization conversion ratio (PCR) of PB meta-atoms illuminated by impinging EM waves linearly polarized along u and v axes, respectively, obtained by FDTD simulations (dot) and experimental measurements (line). (**c**) Sample picture of the meta-atoms for creating slow-wave meta-surface with the following geometric parameters: l=3.5 mm, h=3 mm, $p_{g}=8\;mm$. (**d**) Dispersion relation of the eigen SWs on the slow-wave meta-surface composed of periodic meta-atoms depicted in c. As the patch width changes from l=0 mm to l=3.5 mm, the slope of the dispersion relation gradually decreases.

**Figure 3 nanomaterials-14-01438-f003:**
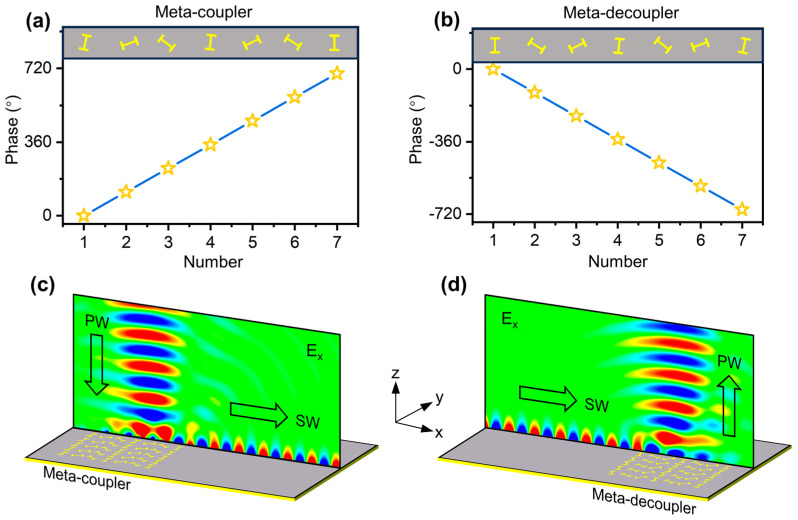
(**a**,**b**) Schematic illustrations and reflection phase distributions of the meta-coupler and meta-decoupler. (**c**,**d**) FDTD simulated E-field (Re(E_x_)) distribution inside the meta-coupler and meta-decoupler under the excitation of PWs and SWs at 12.975 GHz, demonstrating PW-SW conversion and SW-PW decoupling, respectively.

**Figure 4 nanomaterials-14-01438-f004:**
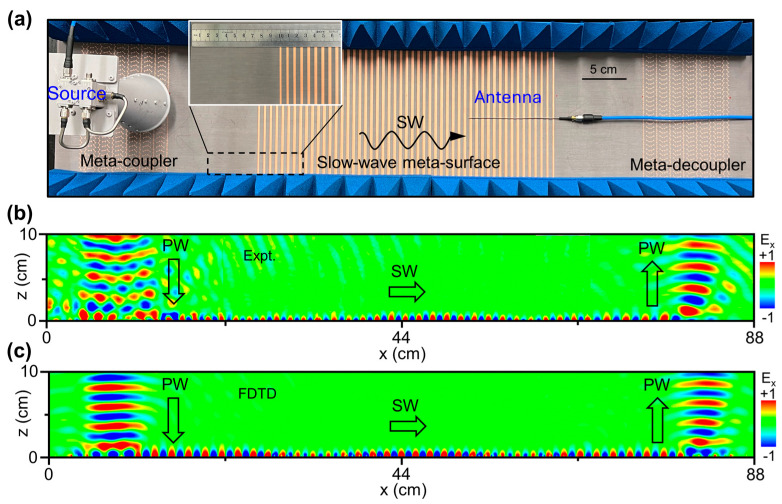
Numerical and experimental demonstration of delaying EM wave effect inside the integrated meta-device. (**a**) Sample picture and near-field measurement of the integrated meta-device, which comprises a meta-coupler, a slow-wave meta-surface, and a meta-decoupler. In the experiment, the CP antenna emits the EM pulse on the left-side meta-coupler, and a monopole antenna is used to detect the electric field distributions (including the amplitude and phase) at the x-z plane. (**b**,**c**) The E-field (Re(E_x_)) distribution on the x-z plane of the integrated meta-device under the illumination of a normally incident Gaussian beam at 12.975 GHz, obtained by the near-field measurement and FDTD simulations, respectively. It is noted that the incident PWs are converted to SWs by the left-side meta-coupler. Finally, the SWs will be radiated back to free space PWs by the right-side meta-decoupler.

**Figure 5 nanomaterials-14-01438-f005:**
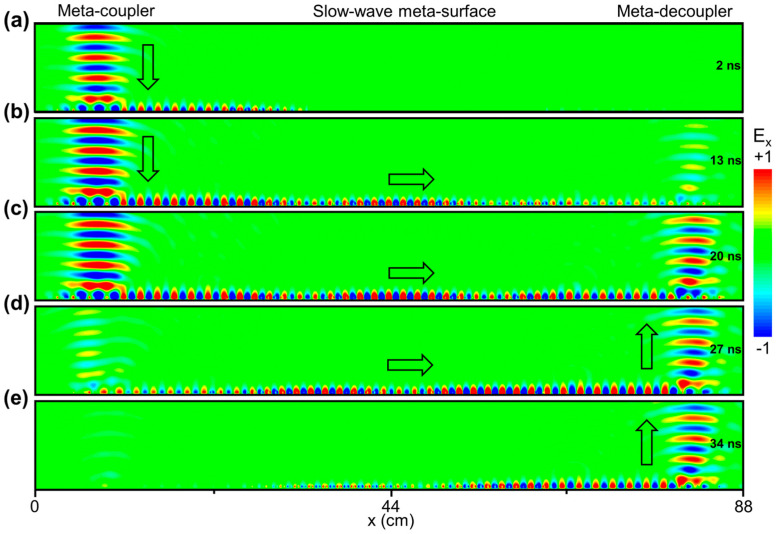
Calculated E-field (Re(E_x_)) distribution inside the meta-device under the illumination of a Gaussian beam with the central frequency of 12.975 GHz at different times through transient simulation. At 2 ns (**a**), a Gaussian PW pulse is just excited by the upper boundary and converted into SW by the left-side meta-coupler. Then, the excited SW will go through the slow-wave meta-surface with the velocity gradually reduced down and then released back, thus accumulating a large delay time (**b**–**e**).

**Figure 6 nanomaterials-14-01438-f006:**
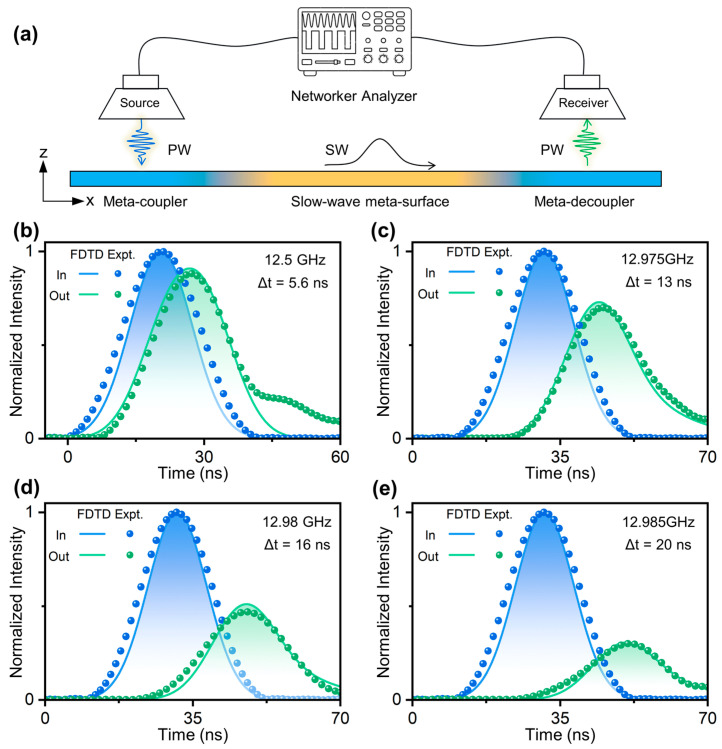
Far-field measurement for evaluating quantitively the delay time of EM pulse achieved by the proposed meta-device. (**a**) Schematic diagram of the far-field measurement setup. In this configuration, the left-side antenna functions as the source, whereas the right-side antenna serves as the receiver. (**b**–**e**) Normalized electric field intensity signals of both the input and output EM pulses with different central frequencies (12.5 GHz, 12. 975 GHz, 12.98 GHz, 12.985 GHz), with their intervals defined as the delay time (5.6 ns, 13 ns, 16 ns, and 20 ns), obtained by FDTD simulations and far-field measurements.

## Data Availability

Data is contained within the article or Appendix A.

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
