# Peer review of "Delaying an Electromagnetic Pulse with a Reflective High-Integration Meta-Platform"

_nanomaterials, 2024, doi:10.3390/nano14171438_

Round 1
Reviewer 1 Report
Comments and Suggestions for Authors
I read with great attention the manuscript 3147060 titled “Delaying electromagnetic pulse with a reflective high-efficiency meta-platform” by Liangwei Li et al that was submitted for publication on Nanomaterials.
The manuscript deals with a technological arrangement to delay pulses that somehow are reflected by the device. I mention somehow because the system does not act like a mirror, but more on a integrated circuit that is excited and deexcited. Indeed the proposed device describes an integrated circuits based on Surface Plasmon Polariton waves: it is constituted by a coupling part, a propagation one and a decoupling. Both coupling and decoupling portions are based on an “innovative grating” based on previous papers that promise 100% efficiency (even if lower efficiencies have been experimentally measured). It must be mentioned that the innovative coupling-decoupling systems have not been experimentally tested in the light band but in the millimetric-centimetric wavelength regions for constructive simplicity (and I am not sure that such geometries can be effectively reproduced in the light band dimensions).
Even if the idea could be interesting, many inaccuracies and omissions are present that make the manuscript unpublishable in the present form. These are the main criticisms found:
Which simulation tool was used? The authors should specify how the simulations were performed.
Did they use 2D or 3D geometries for numerical solution?
Which type of Surface Waves did they excite?
About the 100% input coupler: how can you obtain the interference of the light in the input metacoupler if the cover layer has a perfectly matched dielectric constant? How can you vary continuously the dielectric constant of the cover layer?
At the beginning of pag. 6 the authors introduce the slow-wave metasurface shown in fig.2c. It is not clear at all what fig.2c shows and how the slow-wave metasurface works. They should describe the basis of the slow-light performances.
The manuscript does not report any comparison between slow-wave and no-slow-wave performances. This part must be introduced to show that that their performances are indeed delayed by the metasurface structure.
Figg. 6 b-c-d-e show big losses as function of the delay: the output is strongly decreasing in intensity as the delay increases. The manuscript must report a study on the losses in function of the obtained delay.
The article claims a delay time of 36 ns even if now experimental results are reported at such delay (Figure 6 reports up to 20 ns). Was it really obtained of wished?
Moreover: the article claims a delay time of 36 ns and a conversion efficiency of 73%. Fitting the results reported in figg.6 by a negative exponential trend [(5.6ns;92%) (13ns;70%) (16ns;56%) (20ns;30%)] one obtains (36ns;5.5%). Thus, at 36 ns delay the device efficiency is as low as 5.5% :how can they say 73%?
Fig.5 describes a simulation of the realized device with an overall size of 88cm. The light frequency of 12.975GHz corresponds to a wavelength of about 2.3 cm. Rescaling the device of the wavelength of 1550nm, the overall dimension of the device arrives to about 60 um. Is it a nanometric device?
The overall idea is somehow interesting, but the realization in indeed unclear: why Surface-Plasmon-Polariton waves are needed? I would expect coupling and decoupling of traditional photonic integrated circuits, for example in glass or in Silicon, where you can make the integrated portion as long as needed with much much lower losses. The use of SPP indeed introduces high losses, much higher than the claimed ones.
The manuscript fails for many aspects, and I think it cannot publish on Nanomaterials.
Reviewer 2 Report
Comments and Suggestions for Authors
This paper describes integrated metasurfaces that give a large temporal delay to an electromagnetic pulse. The authors successfully demonstrate a delay of 13 ns at 12.975 GHz with the group velocity of 0.08 times the speed of light in vacuum by both simulations and experiments. It would be of interest to Nanomaterial readers, however, I have some concerns as follows:
1) In page 9, 1st line, Δt=0.1 ns seems too small for the EM pulse shown in Fig. 5.
2) In page 9, last paragraph, the authors should indicate the power efficiency in far-field experiments. How much percentage of sent power was received by the receiver antenna?
3) The authors should mention why the experimental data does not agree with the FDTD simulation after 40 ns in Figure 6b.
4) The authors should also mention why both the simulation and experimental data are not symmetric to the pulse center but have long tail in Figures 6c-e.
5) In page 10, 1st paragraph, the measured delaying time is 13 ns, 16 ns and 20 ns at frequencies of 12.975 GHz, 12.98 GHz, and 12.985 GHz, respectively. It means this system has large delay time dispersion (frequency dependence). Does it cause any problem in practical applications?
Round 2
Reviewer 1 Report
Comments and Suggestions for Authors
I have read the new corrected version of the manuscript. The text has very few changes, and all the clarifications are reported in the supplementary file: here mainly figures are reported and little is explained: however it is quite clear that the delay is related to the integrated structure that slows down the light. In fact, as also reported in figure S5, the delay depends nonlinearly on the frequency of the radiation used.
I understand that the authors want to keep the efficiency high at all costs, but as previously revealed the realistic figure is 30%, that is, maintaining information on the propagation losses that are actually part of the device and absolutely cannot be neglected.
Furthermore, figure S5 points out that the efficiency of 30% is associated with a delay of 10 ns; if you want to reach higher delays, such as the value of 20 ns cited several times in the article, the efficiency drops significantly, reaching a value of 10% or lower.
The authors have certainly done a great job that ultimately deserves to be published: however, I remain of the opinion that the interest of this article is very limited, also because I doubt that it will ever be used and therefore applied in optical regime. The complexity of the structure and the very low efficiency make the device not very interesting and convincing.